# Investigation of Nonlinear Forced Vibrations of the "Rotor-Movable Foundation" System on Rolling Bearings by the Jacobi Elliptic Functions Method

**Almatbek Kydyrbekuly [1,*], Algazy Zhauyt [2] and Gulama-Garip Alisher Ibrayev [1]**

1    Department of Mechanics, Al-Farabi Kazakh National University, Almaty 050040, Kazakhstan; ybraev.alysher@mail.ru
2    Department of Electronics and Robotics, Almaty University of Power Engineering and Telecommunications, Almaty 050013, Kazakhstan; ali84jauit@mail.ru
*    Correspondence: almatbek@list.ru; Tel.: +327-7053256860

**Abstract:** The paper considers a rotor system with a nonlinear characteristic. Its equations of motion are a kind of Duffing class equations with multiple degrees of freedom. The paper shows the advantage of using the method of elliptic functions for solving problems of this type. This method enables us to take into account not only vibrations of the rotor installed in elastic nonlinear supports, but also vibrations of the foundation. A comparative analysis of application of the method of elliptic functions proposed by the authors is carried out by comparing the derived equations of motion of the system, as well as by comparing the obtained amplitude-frequency characteristics with the results obtained by the numerical Runge–Kutta–Fehlberg's 4-order method and the approximate analytical Van der Pol method. The regions of resonant frequencies for superharmonic oscillations and bifurcation regimes are determined. It is concluded that the method proposed by the authors is a more accurate and general case than the previously used approximate methods.

**Keywords:** nonlinear dynamics of rotor systems; steady-state forced oscillations; elliptic functions; duffing equation

## 1. Introduction

A significant contribution to the study of linear and nonlinear dynamics of rotor systems was made by such scientists as Rao [1], Yamamoto and Ishida [2], Tondle [3], Mushinskaya [4], Jeffcott [5], Dimentberg [6], Stodola [7], Kelzon [8], Tiwari [9], Cramer [10], Adams [11], Vance [12], Penny [13], and others, whose works are the basis of those physical and mathematical concepts that contributed to its further development. One of the main components determining the reliability of rotary machines is elastic supports [14–17]. In this paper, rolling bearings with a non-linear characteristic act as elastic supports. The neglect of nonlinear properties in the study of the dynamics of vibrations of mechanical systems usually leads to overestimated amplitudes and frequencies, which, accordingly, worsens qualitative and quantitative results, and causes unnecessary additional costs in the design of rotor systems [18–20]. In the linear theory of the dynamics of rotary machines, damping and elastic properties of bearings are not accurately specified, which is especially important when it is necessary to study high-speed and high-frequency processes. In fact, the rigidity of a rolling bearing depends on many factors, among which one can note the size of the gap and its geometry [21], the type of load and the mode of operation of the system [22–25], the number of rolling elements in the bearing, the size and type of fit of the rings, etc. [24,26–28]. Often, Hertz's contact theory is used to analyze and describe the restoring forces of a rolling bearing. It connects the deformation at the points of contact between the rolling element of the bearing with the loads acting on the bearing in the radial direction, without taking into account the slippage of the rolling elements

or rolling surfaces [19,20,28]. To describe the nonlinear properties of the restoring force of rolling bearings, the cubic power series approximation [4,18,28,29] is used, since this approximation is in good agreement over a wide range of deformation.

In addition, for the most complete analysis of the process, it is important to take into account the influence of such factors as the asymmetry of the installation of the rotor on the shaft, i.e., imbalance, variability of inertial parameters, various positional forces, external friction, mobility of the foundation, etc. The impact of foundation vibrations on the linear dynamics of the "rotor-bearings" system and, in particular, on rotor vibrations was studied in detail in [5,6,10,11,30,31].

In Refs. [30,31], mathematical models of the "rotor-bearing-foundation" system, in a linear formulation, with the rotor shaft located horizontally, are presented. In a nonlinear formulation, this problem was partially considered and analyzed by the method of complex amplitudes in [32,33]. A more complicated mathematical model of this kind makes it possible to investigate the effect of rotational speed on frequency spectra and amplitude-frequency characteristics for any rotor system on rolling bearings.

In this paper, the motion of the rotor system is described by a system of cubically non-linear non-homogeneous differential equations of the second order, known as the Duffing equations. In most cases, nonlinear differential equations are divided into equations with "weak" and "strong" nonlinearity according to the value of the nonlinearity parameter. The case with "weak" nonlinearity allows an approximate solution of the equation, since, as practice shows, the solution in this case is almost periodic. The case of eigenoscillations has been fairly well studied with sufficient accuracy by many authors using various methods, and the case of forced oscillations has been less studied [34].

It should be noted that there are also works on finding exact solutions to the Duffing equation for some isolated cases, expressed in terms of elliptic functions, described in the works of such authors as Tsvetitsanin, Kovacic, Hsu [35,36], and Starossek [37]. The solution of the Duffing equation using the method of elliptic functions was first shown by Hsu in [36]. On the basis of elliptic functions, the approximate methods of Poincare–Lindstedt, Krylov–Bogolyubov, the generalized Galerkin method, the multiple scales method, and the harmonic balance method were developed [35]. An analysis of the sources allows us to conclude that the method of elliptic functions has many advantages when searching for both approximate and exact solutions of quadratic oscillators, Duffing, and Helmholtz oscillators [34,35]. The method of Jacobi elliptic functions is also used in the study of rotor systems, in particular, in [38], where the Jeffcot rotor is considered. A review of the literature for the period of over 20 years shows that there are practically no theoretical works on the dynamics of vertical rotor systems in this formulation, where the method of investigation is the method of elliptic functions and the corresponding experimental works. In this regard, in this paper, for the first time, the system "Rotor-Rolling Bearings with a Nonlinear Characteristic-Movable Foundation" is investigated, which, as will be shown below, allows us to obtain more generalized results based on the method of elliptic functions.

## 2. Problem Statement

A symmetrical vertical rotor of mass $m$, with polar moment of inertia $J$, having a static imbalance $e$, rotates on rolling bearings with a constant angular velocity $\Omega_0$. The foundation of mass $M$ is installed on elastic supports with an equivalent linear stiffness coefficient $c_2$ (Figure 1). It is assumed that the rotor performs a plane-parallel motion, and there is no rotation around the coordinate axes. The radial compliance of bearings occurs due to the deformation of the rolling elements and raceways at the points of contact.

Two Cartesian coordinate systems are introduced, fixed—$Oxyz$ and mobile, rigidly connected with the geometric center of the rotor—$O_1\xi\eta$, where $\eta$ is the polar axis, and the $\xi$ axis is arbitrarily drawn through the eccentricity vector of the rotor mass. The motion of the system is considered relative to the fixed coordinate system $Oxyz$. The coordinates of the geometric center of the rotor are designated as $O_1(x_1, y_1)$, and its center of mass as

$O_s(x, y)$. The foundation mass center coordinates are denoted as $O_2(x_2, y_2)$, $\chi$ and $\chi_0$ are damping coefficients (Figure 1).

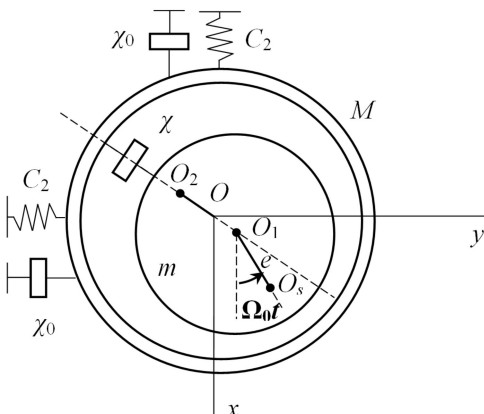

**Figure 1.** Rotor system on rolling bearings.

The non-linear restoring force of a bearing can be described using the Hertzian contact formula:

$$F_c = C_b \delta_r^{\frac{3}{2}} \tag{1}$$

where $F_C$ is the radial force, $\delta_r$ is the radial deformation, $C_b$ is the non-linear stiffness coefficient.

For a qualitative analysis of the equations of motion of the system, the restoring force of rolling bearings in the form (1) can be approximated by a power series:

$$F_c = c_0 \delta_r + c_1 \delta_r^3, \tag{2}$$

where $c_0$ and $c_1$ are cubic and linear coefficients of a cubic polynomial, respectively. The kinetic energy of the system is defined as:

$$T = \frac{1}{2}\left(m\left(\dot{x}^2 + \dot{y}^2\right) + J\Omega_0^2 + M\left(\dot{x}_2^2 + \dot{y}_2^2\right)\right). \tag{3}$$

Considering that:

$$\delta_r^2 = (x_1 - x_2)^2 + (y_1 - y_2)^2. \tag{4}$$

The potential energy of the isotropic non-linear elastic field of the system is defined as:

$$\Pi = \frac{1}{2}\left(c_0\left((x_1 - x_2)^2 + (y_1 - y_2)^2\right) + c_2\left(x_2^2 + y_2^2\right)\right) + \frac{1}{4}\left(c_1\left((x_1 - x_2)^4 + (y_1 - y_2)^4\right)\right). \tag{5}$$

The energy dissipation function reads:

$$\Phi = \frac{1}{2}\left(\chi\left(\dot{x}^2 + \dot{y}^2\right) + \chi_0\left(\dot{x}_2^2 + \dot{y}_2^2\right)\right). \tag{6}$$

The coordinates of the center of mass of the rotor are given as:

$$x = x_1 + e\cos\Omega_0 t, y = y_1 + e\sin\Omega_0 t, \tag{7}$$

where $e$ is the value of the linear displacement of the center of mass of the rotor from its geometric center $O_1 O_s$, i.e., static imbalance. Using the Lagrange equations of the second kind, we write the equations of motion of the system in the following form:

$$
\begin{aligned}
m\ddot{x}_1 + c_0(x_1 - x_2) + c_1(x_1 - x_2)^3 + \chi \dot{x}_1 &= me\Omega_0^2 \cos \Omega_0 t, \\
m\ddot{y}_1 + c_0(y_1 - y_2) + c_1(y_1 - y_2)^3 + \chi \dot{y}_1 &= me\Omega_0^2 \sin \Omega_0 t, \\
M\ddot{x}_2 - c_0(x_1 - x_2) - c_1(x_1 - x_2)^3 + c_2 x_2 + \chi_0 \dot{x}_2 &= 0, \\
M\ddot{y}_2 - c_0(y_1 - y_2) - c_1(y_1 - y_2)^3 + c_2 y_2 + \chi_0 \dot{y}_2 &= 0, \\
x_1(0) = e, x_2(0) = 0.1e, y_1(0) &= 0, y_2(0) = 0, \\
\dot{x}_1(0) = 0, \dot{x}_2(0) = 0, \dot{y}_1(0) &= 0, \dot{y}_2(0) = 0.
\end{aligned}
\tag{8}
$$

Equation (8) describes joint movement of the movable foundation and the rotor with static imbalance installed in elastic supports with a nonlinear characteristic. In the case of a fixed foundation, Equation (8) will consist of two Duffing equations describing the equation of motion of a nonlinear Jeffcot rotor, which have been studied in sufficient detail.

We introduce the following dimensionless parameters:

$$
\begin{aligned}
x_1 &= ef_1, x_2 = ef_2, y_1 = ev_1, y_2 = ev_2, \\
\mu &= \tfrac{m}{M}, \omega_1^2 = \tfrac{c_0}{m}, \omega_2^2 = \tfrac{c_2}{m}, \tau = \omega_1 t, \ \Omega_0 = \omega_1 \eta, \\
\zeta_1 &= \tfrac{\chi}{2m\omega_1}, \zeta_2 = \tfrac{\chi_0}{2m\omega_1}, \varepsilon = \tfrac{c_1 e^2}{m\omega_1^2}, \lambda = \tfrac{\omega_2^2}{\omega_1^2}.
\end{aligned}
\tag{9}
$$

Then, through the system of Equation (8), we obtain:

$$
\begin{aligned}
\ddot{f}_1 + 2\zeta_1 \dot{f}_1 + (f_1 - f_2) + \varepsilon(f_1 - f_2)^3 &= \eta^2 \cos(\eta\tau), \\
\ddot{v}_1 + 2\zeta_1 \dot{v}_1 + (v_1 - v_2) + \varepsilon(v_1 - v_2)^3 &= \eta^2 \sin(\eta\tau), \\
\ddot{f}_2 + 2\mu\zeta_2 \dot{f}_2 - \mu(f_1 - f_2) - \mu\varepsilon(f_1 - f_2)^3 + \mu\lambda f_2 &= 0, \\
\ddot{v}_2 + 2\mu\zeta_2 \dot{v}_1 - \mu(v_1 - v_2) - \mu\varepsilon(v_1 - v_2)^3 + \mu\lambda v_2 &= 0.
\end{aligned}
\tag{10}
$$

*Equations of Motion's Solution*

Since we are looking for a quasi-periodic solution of Equation (10) and, accordingly, we consider the case with weak nonlinearity, we will use the classic proven Van der Pol method to compare the results that will be obtained using the Jacobi elliptic function method. Since there are damping forces in the system, we represent the functions $f_1(\tau)$ and $f_2(\tau)$ in system (10) in the following form:

$$
\begin{aligned}
f_1(\tau) &= A_1(\tau) \cos(\eta\tau) + B_1(\tau) \sin(\eta\tau), \\
f_2(\tau) &= A_2(\tau) \cos(\eta\tau) + B_2(\tau) \sin(\eta\tau),
\end{aligned}
\tag{11}
$$

where derivatives $A_1(\tau)$, $A_2(\tau)$, $B_1(\tau)$, and $B_2(\tau)$ are amplitudes slowly changing over the oscillation period. Substituting (11) into the Equation (10) and equating the left and right parts of the equations for the same functions and harmonics, we obtain:

$$
\begin{aligned}
\ddot{A}_1 - \eta^2 A_1 + 2\dot{B}_1\eta + 2\zeta_1\left(\dot{A}_1 + B_1\eta\right) + (A_1 - A_2) + \tfrac{3}{4}\varepsilon\left((A_1 - A_2)^3 + (A_1 - A_2)(B_1 - B_2)^2\right) &= \eta^2, \\
\ddot{B}_1 - \eta^2 B_1 - 2\dot{A}_1\eta + 2\zeta_1\left(\dot{B}_1 - A_1\eta\right) + (B_1 - B_2) + \tfrac{3}{4}\varepsilon\left((B_1 - B_2)^3 + (A_1 - A_2)^2(B_1 - B_2)\right) &= 0, \\
\ddot{A}_2 - \eta^2 A_2 + 2\dot{B}_2\eta + 2\mu\zeta_2\left(\dot{A}_2 + B_2\eta\right) - \mu(A_1 - A_2) - \tfrac{3}{4}\mu\varepsilon\left((A_1 - A_2)^3 + (A_1 - A_2)(B_1 - B_2)^2\right) + \mu\lambda A_2 &= 0, \\
\ddot{B}_2 - \eta^2 B_2 - 2\dot{A}_2\eta + 2\mu\zeta_1\left(\dot{B}_2 - A_2\eta\right) - \mu(B_1 - B_2) - \tfrac{3}{4}\mu\varepsilon\left((B_1 - B_2)^3 + (A_1 - A_2)^2(B_1 - B_2)\right) + \mu\lambda B_2 &= 0.
\end{aligned}
\tag{12}
$$

The first and second order derivatives of the amplitudes of the rotor and the foundation are equal to zero, since for the stationarity of the solutions of system (12) it is necessary that $A_1(\tau)$, $A_2(\tau)$, $B_1(\tau)$), and $B_2(\tau)$ be constant in time, then:

$$(1 - \eta^2)A_1 - A_2 + 2\eta\zeta_1 B_1 + \tfrac{3}{4}\varepsilon\Big((A_1 - A_2)^3 + (A_1 - A_2)(B_1 - B_2)^2\Big) = \eta^2,$$
$$(1 - \eta^2)B_1 - B_2 + 2\eta\zeta_1 A_1 + \tfrac{3}{4}\varepsilon\Big((B_1 - B_2)^3 + (A_1 - A_2)^2(B_1 - B_2)\Big) = 0,$$
$$-\mu A_1 + (\mu\lambda - \eta^2 + \mu)A_2 + 2\mu\eta\zeta_2 B_2 - \tfrac{3}{4}\mu\varepsilon\Big((A_1 - A_2)^3 + (A_1 - A_2)(B_1 - B_2)^2\Big) = 0,$$
$$-\mu B_1 + (\mu\lambda - \eta^2 + \mu)B_2 + 2\mu\eta\zeta_2 A_2 - \tfrac{3}{4}\mu\varepsilon\Big((A_1 - A_2)^3 + (A_1 - A_2)(B_1 - B_2)^2\Big) = 0. \tag{13}$$

From the algebraic equations of system (13), by varying the dimensionless frequency $\eta$, and analytically solving cubic amplitude polynomials for each case, the amplitude-frequency characteristics, shown in figures below, were obtained.

As the elliptic functions are periodic functions on the complex plane, they can be considered as a generalized case of trigonometric functions. In the case of stable forced oscillations, the oscillation trajectory of the center of inertia of a linear oscillator can be depicted as a circle (Figure 2, case (a)). In the case of small nonlinearity, the trajectory of the center of inertia of our system, which is a nonlinear oscillator, will describe an ellipse (Figure 2, case (b)). Therefore, geometrically elliptic functions can be represented as:

$$\mathrm{cn}(u,k) = \frac{x}{a}, \mathrm{sn}(u,k) = \frac{y}{b}, \mathrm{dn}(u,k) = \frac{r}{a}, \tag{14}$$

where $b = 1$, and:

$$u = \int_A^B r\,d\varphi = \int_A^B \frac{d\varphi}{\sqrt{1 - k^2 \sin^2 \varphi}}, k = \sqrt{1 - \frac{1}{a^2}}, \tag{15}$$

where

$$\varphi = amp(u,k) = \int_0^u \mathrm{dn}(u',k)\,du', \tag{16}$$

here $amp(u)$ is an inverse function of the elliptic integral of the first kind:

$$u = F(\varphi,k) = \int_0^\varphi \frac{d\xi}{\sqrt{1 - k^2 \sin^2 \xi}}. \tag{17}$$

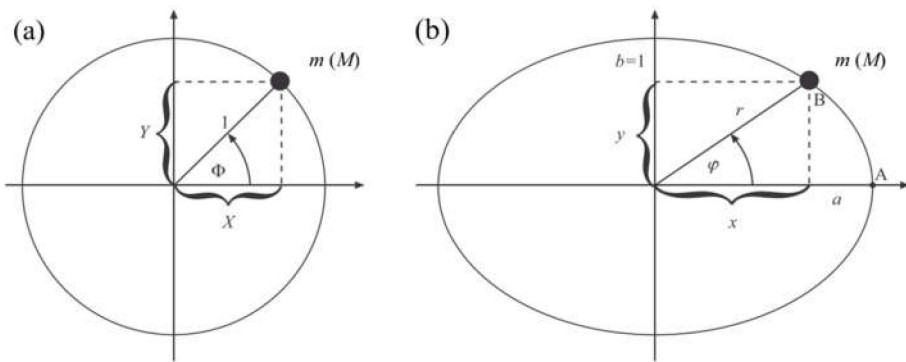

**Figure 2.** Geometrical representation of the oscillatory motion of the center of inertia of the rotor and the foundation: (**a**)—Linear Case, (**b**)—Nonlinear Case.

Taking into account the above expressions (14)–(17), we represent in the form (18) the main relations for elliptic functions used in the following sections in the form:

$$\text{cn}^2(u,k) + \text{sn}^2(u,k) = 1, \text{dn}^2(u,k) + k^2\text{sn}^2(u,k) = 1, \text{dn}(u,k) = \sqrt{1 - k^2\text{sn}^2(u,k)},$$
$$\tfrac{d}{du}\text{sn}(u,k) = \text{cn}(u,k)\text{dn}(u,k), \tfrac{d}{du}\text{cn}(u,k) = -\text{sn}(u,k)\text{dn}(u,k), \tag{18}$$
$$\tfrac{d}{du}\text{dn}(u,k) = -k^2\text{sn}(u,k)\text{cn}(u,k).$$

For a more accurate determination of the amplitudes and resonant frequencies of forced oscillations, we use the Jacobi elliptic functions. To solve Equation (10) by the elliptic function method, we assume that the perturbing force $\cos(\eta\tau)$ is a special case of the elliptic cosine $\text{cn}(\omega_f\tau,k)$, i.e., we assume that $\eta \equiv \omega_f$, and for $k = 0$, $\text{cn}(\omega_f\tau, 0) = \cos(\eta\tau)$. Taking these statements into account, we rewrite system (10) in the following general form:

$$\ddot{f}_1 + 2\zeta_1\dot{f}_1 + (f_1 - f_2) + \varepsilon(f_1 - f_2)^3 = \omega_f^2\text{cn}(\omega_f\tau,k),$$
$$\ddot{f}_2 + 2\mu\zeta_2\dot{f}_2 - \mu(f_1 - f_2) - \mu\varepsilon(f_1 - f_2)^3 + \mu\lambda f_2 = 0, \tag{19}$$

where $\omega_f$ is a circular frequency of the perturbing force, $k$ is the elliptic modulus, and $\text{cn}(\omega_f\tau,k)$ and $\text{sn}(\omega_f\tau,k)$ are the elliptic cosine and sine.

The solution to Equation (19), as damping forces are present in the system, will be sought in the following form:

$$f_1(\tau) = A_1\text{cn}(\omega_f\tau,k) + B_1\text{sn}(\omega_f\tau,k),$$
$$f_2(\tau) = A_2\text{cn}(\omega_f\tau,k) + B_2\text{sn}(\omega_f\tau,k), \tag{20}$$

Taking into account the main relations of elliptic functions, we obtain:

$$\tfrac{d}{d\tau}f_i = \tfrac{df_i}{du}\tfrac{du}{d\tau} = -\omega_f A_i\text{sndn} + \omega_f B_i\text{cndn},$$
$$\tfrac{d^2}{d\tau^2}f_i = -\omega_f^2 A_i(1 - 2k^2\text{sn}^2) - \omega_f^2 B_i(1 - k^2 + 2k^2\text{cn}^2), \tag{21}$$

where $i = 1, 2$; $\text{cn} \equiv \text{cn}(\omega_f\tau,k)$, $\text{sn} \equiv \text{sn}(\omega_f\tau,k)$, $\text{dn} \equiv \text{dn}(\omega_f\tau,k)$.

Substituting the solution in the form (20) into system (19), and, taking into account (21), we obtain:

$$\text{cn}\Big(\omega_f^2(2k^2 - 1)A_1 \quad +(A_1 - A_2) + 3\varepsilon(A_1 - A_2)(B_1 - B_2)^2 - \omega_f^2 + 2\omega_f\zeta_1 B_1 H_1\Big)$$
$$+\text{sn}\Big(\omega_f^2(k^2 - 1)B_1 + (B_1 - B_2) + 3\varepsilon(B_1 - B_2)^3 - 2\omega_f\zeta_1 A_1 H_1$$
$$+H_3\Big(\varepsilon\Big(3(A_1 - A_2)^2 - (B_1 - B_2)^2\Big)(B_1 - B_2) - 2\omega_f^2 k^2 B_1\Big)\Big)$$
$$+\text{cn}^3\Big((A_1 - A_2)^3 - 2\omega_f^2 k^2 A_1 - 3\varepsilon(A_1 - A_2)(B_1 - B_2)^2\Big) = 0,$$
$$\text{cn}\Big(\omega_f^2(2k^2 - 1)A_2 \quad -\mu(A_1 - A_2) - 3\mu\varepsilon(A_1 - A_2)(B_1 - B_2)^2 + \mu\lambda A_2 + 2\omega_f\zeta_2 B_2 H_2\Big) \tag{22}$$
$$+\text{sn}\Big(\omega_f^2(k^2 - 1)B_2 - \mu(B_1 - B_2) - \mu\varepsilon(B_1 - B_2)^3 + \mu\lambda B_2 - 2\mu\omega_f\zeta_2 A_2 H_1$$
$$+H_3\Big(-\mu\varepsilon\Big(3(A_1 - A_2)^2 - (B_1 - B_2)^2\Big)(B_1 - B_2) - 2\omega_f^2 k^2 B_2\Big)\Big)$$
$$+\text{cn}^3\Big(-\mu\varepsilon(A_1 - A_2)^3 - 2\omega_f^2 k^2 A_2 + 3\mu\varepsilon(A_1 - A_2)(B_1 - B_2)^2\Big) = 0,$$

where is:

$$H_1 = \frac{1}{\pi} \int\limits_0^{2\pi} \text{sn} u \text{dn} u \sin \varphi d\varphi = \frac{1}{\pi} \int\limits_0^{4K} \text{sn}^2 u \text{dn}^2 u du = \frac{4}{3k^2\pi}\left((2k^2-1)E + K(1-k^2)\right),$$

$$H_2 = \frac{1}{\pi} \int\limits_0^{2\pi} \text{cn} u \text{dn} u \cos \varphi d\varphi = \frac{1}{\pi} \int\limits_0^{4K} \text{cn}^2 u \text{dn}^2 u du = \frac{4}{3k^2\pi}\left((1+k^2)E - K(1-k^2)\right),$$

$$H_3 = \frac{1}{\pi} \int\limits_0^{2\pi} \text{sn} u \text{cn}^2 u \sin \varphi d\varphi = \frac{1}{\pi} \int\limits_0^{4K} \text{sn}^2 u \text{cn}^2 u \text{dn} u du = \frac{1}{4},$$

$$K = \int\limits_0^{\varphi} \frac{d\varphi}{\sqrt{1-k^2\sin^2\varphi}} \approx \frac{\pi}{2}\left(1 + \frac{1}{4}k^2 + \frac{9}{64}k^4 + O(k^6)\right),$$

$$E = \int\limits_0^{\varphi} \sqrt{1-k^2\sin^2\varphi}d\varphi \approx \frac{\pi}{2}\left(1 - \frac{1}{4}k^2 - \frac{3}{64}k^4 + O(k^6)\right).$$

Here $K \equiv K(\varphi, k)$ is a complete elliptic integral of the first kind, whereas $E \equiv E(\varphi, k)$ is an incomplete elliptic integral of the second kind.

Equating the coefficients for the same functions and harmonics of Equation (22), we obtain a system of algebraic equations that enables us to determine the amplitudes of forced vibrations in the form (23):

$$(A_1 - A_2) + \frac{1}{2}\omega_f^2 k^2 A_1 - \omega_f^2 A_1 + \frac{3}{4}\varepsilon(A_1 - A_2)\left((A_1 - A_2)^2 + (B_1 - B_2)^2\right) - \omega_f^2 + 2\omega_f\zeta_1 B_1 H_2 = 0,$$

$$(B_1 - B_2) + \frac{1}{2}\omega_f^2 k^2 B_1 - \omega_f^2 B_1 + \frac{3}{4}\varepsilon(A_1 - A_2)\left((A_1 - A_2)^2 + (B_1 - B_2)^2\right) - 2\omega_f\zeta_1 A_1 H_1 = 0,$$

$$\varepsilon(A_1 - A_2)^3 - 2\omega_f^2 k^2 A_1 - 3\varepsilon(A_1 - A_2)(B_1 - B_2)^2 = 0,$$

$$\mu((\lambda+1)A_2 - A_1) + \frac{1}{2}\omega_f^2 k^2 A_2 - \omega_f^2 A_2 - \frac{3}{4}\mu\varepsilon(A_1 - A_2)\left((A_1 - A_2)^2 + (B_1 - B_2)^2\right) + 2\omega_f\zeta_2 B_2 H_2 = 0,$$

$$\mu((\lambda+1)B_2 - B_1) + \frac{1}{2}\omega_f^2 k^2 B_2 - \omega_f^2 B_2 - \frac{3}{4}\mu\varepsilon(B_1 - B_2)\left((A_1 - A_2)^2 + (B_1 - B_2)^2\right) - 2\omega_f\zeta_2 A_2 H_1 = 0,$$ (23)

$$-\mu\varepsilon(A_1 - A_2)^3 - 2\omega_f^2 k^2 A_2 + 3\mu\varepsilon(A_1 - A_2)(B_1 - B_2)^2 = 0,$$

$$k^2 = \frac{\varepsilon(A_1 - A_2)\left((A_1 - A_2)^2 - 3(B_1 - B_2)^2\right)}{2\omega_f^2 A_1}.$$

By varying the cyclic frequency in the system of Equation (23), the amplitude-frequency characteristics of the forced oscillations of the rotor and foundation were obtained (Figures 3–12). As can be seen, for $k = 0$, the system of Equation (23) degenerates into system (13). Equation (13) can be obtained by such classical approximate methods as the Van der Pol method, the harmonic balance method, etc. This fact is a direct confirmation of the generalization of the method of elliptic functions.

## 3. Results

The amplitude-frequency characteristics for an industrial centrifuge with parameters $m = 24$ kg, $M = 25$ kg, $e = 0.001$ m, $c_0 = 1.1 \times 10^7$ kg/s$^2$, $c_1 = 0.87 \times 10^{11}$ kg/m$^2$s$^2$, $c_2 = 3.26 \times 10^5$ kg/s$^2$, $\chi_0 = 6.59$ kg/s, $\chi = 4200$ kg/s were obtained by the following methods: from the algebraic system of Equation (13) using the Van der Pol's method (Figures 3 and 4, the blue curve), from differential Equation (8) using the 4th order Runge–Kutta–Fehlberg method (Figures 3 and 4, the dotted red curve), from differential Equation (8) at $c_1 = 0$ by the 4th order Runge–Kutta–Fehlberg method (Figures 3 and 4, the black curve) and compared with the amplitude-frequency characteristic obtained from the algebraic system of Equation (23) derived by the method of Jacobi elliptic functions (Figures 3 and 4, the green curve). As a result of calculations, three resonant regions were determined for the rotor and the foundation, the presence of which, as will be shown below, is due to the nonlinearity of the supports and the mobility of the foundation.

The curve of the amplitude-frequency characteristic in the first resonant region is typical of linear oscillations and is observed for all cases at $\omega_f = 0.051$. When calculating the amplitudes by the Van der Pol method (Figures 3 and 4, the blue curve), the deviation from the results of the numerical method (Figures 3 and 4, the dotted red curve) in the case of the rotor was 2.8%, and in the case of the foundation it was 9.1%. In cases of calculation

by the method of elliptic functions, the amplitudes for the first resonance region are the closest to the results of the numerical method. The deviation of the amplitude values for the rotor was 1.14%, and for the foundation 6.4% (Figures 3 and 4, the green curve). The amplitude values are presented in Table 1.

**Table 1.** Amplitude values at the first resonance for different methods.

| Method | First Resonant Frequency-$\omega_f$ | Rotor's Amplitude-$f_1$ | Foundation's Amplitude-$f_2$ |
|---|---|---|---|
| Numerical Solution (Nonlinear Case) | | 1.752 | 3.742 |
| Van der Pol's method | 0.051 | 1.703 | 3.455 |
| Jacobi Elliptic Function method | | 1.732 | 3.564 |

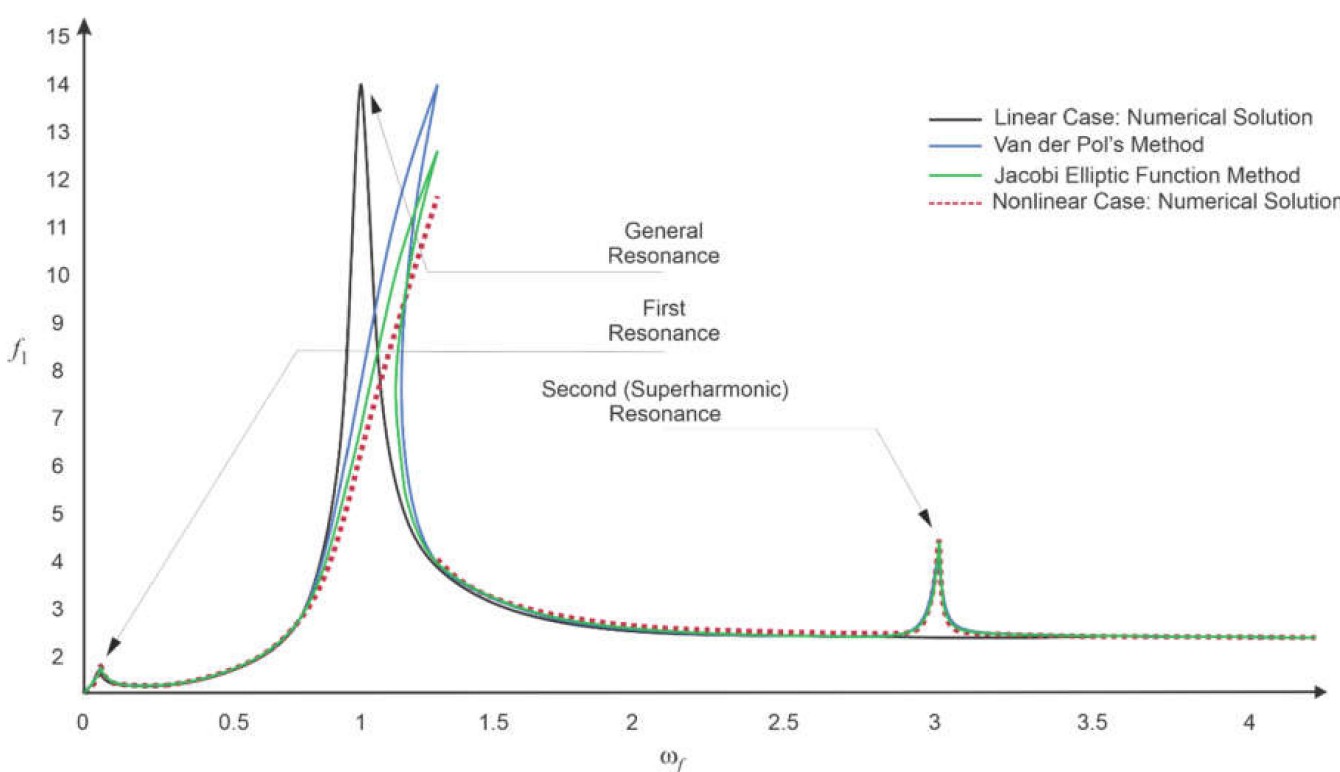

**Figure 3.** The amplitude-frequency characteristic of the rotor.

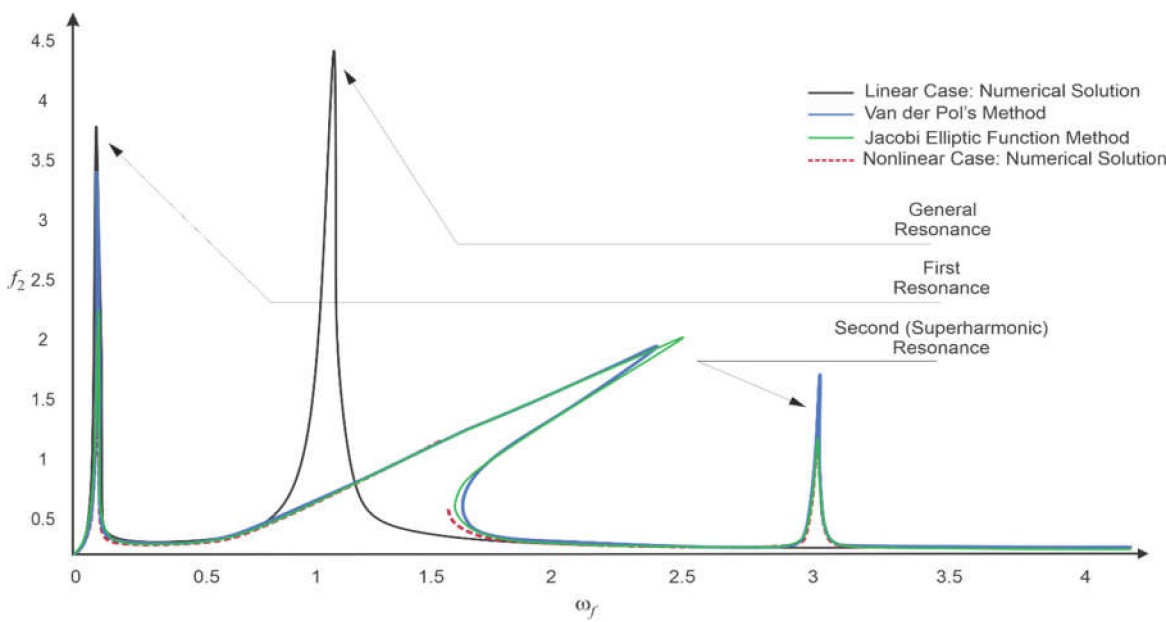

**Figure 4.** The amplitude-frequency characteristic of the foundation.

The amplitude-frequency characteristic curve for the second resonant region is typical of nonlinear oscillations with a rigid characteristic. Nonlinear oscillations and bifurcation modes in this case are due to nonlinearity of the restoring forces of the elastic supports. In this case, the amplitude breakdown for each method occurs at different frequencies with different amplitudes. For example, for the 4th order Runge–Kutta–Fehlberg numerical method, the resonance was observed at $\omega_f = 1.33$, in the case of the Van der Pol method, the deviation from the amplitude of the numerical method for the rotor was 17.6%, for the foundation 33.87%, while the resonance occurred at $\omega_f = 1.25$. In the case of Jacobi elliptic functions, this breakdown can be observed at $\omega_f = 1.3$, the deviation from the amplitudes for the rotor is 8.66%, and for the foundation it is 31.7%. As can be seen in the latter case, the deviations of the values of the resonant frequency and amplitude from the numerical method are much smaller. Amplitude values are presented in Table 2.

**Table 2.** Amplitude values at the second resonance for different methods.

| Method | Second Resonant Frequency-$\omega_f$ | Rotor's Amplitude-$f_1$ | Foundation's Amplitude-$f_2$ |
|---|---|---|---|
| Numerical Solution (Nonlinear Case) | 1.333 | 11.556 | 1.333 |
| Van der Pol's method | 1.256 | 14.022 | 2.016 |
| Jacobi Elliptic Function method | 1.301 | 12.554 | 1.954 |

Nonlinear systems are characterized by anharmonicity, i.e., the presence of ultra-harmonic and subharmonic resonances that are multiples of the frequency of the main resonance. For the linear case, the main resonance occurs at $\omega_f = 1$. In our case, the main resonance region, due to nonlinearity, is observed at $1 \le \omega_f \le 1.33$. In order to determine additional resonances, the amplitude-frequency characteristics of the corresponding linear case (Figures 3 and 4, the black curve), the amplitude-frequency characteristics obtained by the numerical method (Figures 3 and 4, the red curve), and the amplitude-frequency characteristics of the analytical by the Van der Pol method (Figures 3 and 4, the blue curve) were constructed. Since the second resonance, located to the right of the main one at $\omega_f = 1$,

is not observed in the linear case and is a multiple of the main one, it was found that it is an ultraharmonic resonance.

It should be noted that in all the above cases, the main disadvantage of using the numerical method for the nonlinear case is the impossibility of determining the amplitudes of the rotor and the foundation for the amplitudes lying behind the break point. In the case of using the Van der Pol method, large deviations are observed both for the amplitudes and for the resonant frequencies. Jacobi's method of elliptic functions is free from these shortcomings due to its generality.

To optimize the workflow using the method of elliptic functions, a parametric analysis was carried out by constructing the amplitude-frequency characteristics of the rotor and foundation for different values of the nonlinearity parameter $\varepsilon$ (Figures 5 and 6), for different values of the damping coefficients $\zeta_1$ and $\zeta_2$ (Figures 7 and 8), for different values of the linear stiffness parameter $\lambda$ (Figures 9 and 10), and for different values of the mass ratio of the foundation and the rotor $\mu$ (Figures 11 and 12). All variable parameters are reduced to dimensionless form according to (9).

An increase in the nonlinearity parameter up to one order of magnitude inclusive (Figures 5 and 6: cases $\varepsilon = 5$ and $\varepsilon = 10$) leads to a decrease in the amplitudes of the rotor and foundation in the region of the first resonance. For example, for $\varepsilon = 5$, the rotor amplitude decreases by a factor of 1.26, while the foundation amplitude decreases by a factor of 1.08. Further, with an increase in the nonlinearity parameter, i.e., for $\varepsilon = 10$, the decrease in amplitudes is also more intense. For the rotor, in this case, the amplitude relative to $\varepsilon = 1$ already decreases by a factor of 2.27, and for the foundation, by a factor of 1.4. With a decrease in the nonlinearity parameter within one order of magnitude, the opposite picture is observed (Figures 5 and 6: cases $\varepsilon = 0.1$ and $\varepsilon = 0.5$). For example, for $\varepsilon = 0.5$, the rotor amplitude increases by 1.5 times, and the foundation amplitude by 1.13 times. In the case of a decrease in the nonlinearity parameter by one order of magnitude, i.e., when $\varepsilon = 0.1$, the increase in amplitudes is more intense and exceeds the initial values by a factor of 1.96 for the rotor and by a factor of 1.43 for the foundation. No shifts in the region of the first resonant frequencies are observed in these cases. The absence of a shift along the frequency axis of the first resonance is due to the linear nature of the oscillations in this region. In addition, as shown earlier, these resonant amplitudes arise solely due to mobility and oscillations of the foundation. In the region of nonlinear resonant frequencies, the magnitudes of the amplitudes do not depend on the variation of the nonlinearity parameter. With an increase in nonlinearity, the slope of the amplitude-frequency characteristic curve increases, and the amplitude breakdown point shifts in the direction of higher frequencies. For example, for $\varepsilon = 5$, the drop in amplitudes shifts to $\omega_f = 1.4$, whereas for $\varepsilon = 10$, a drop is already observed at $\omega_f = 1.43$. As the nonlinearity decreases, the slope of the curve correspondingly decreases, degenerating into a linear case. The frequency and amplitude values are given in more detail in Table 3 and in Figures 5 and 6. Variation of the non-linearity parameter only affects the slope at the second resonance and the amplitude values, since the non-linearity is known to limit the magnitude of the amplitudes.

**Table 3.** Amplitudes and resonant frequencies for different degrees of non-linearity.

| Parameter Value | First Resonance | | | Second Resonance | | |
|---|---|---|---|---|---|---|
| | Frequency -$\omega_f$ | Rotor's Amplitude-$f_1$ | Foundation's Amplitude-$f_2$ | Frequency -$\omega_f$ | Rotor's Amplitude-$f_1$ | Foundation's Amplitude-$f_2$ |
| $\varepsilon = 0.1$ | | 9.8 | 5 | 1.15 | | |
| $\varepsilon = 0.5$ | | 8 | 3.95 | 1.2 | | |
| $\varepsilon = 1$ | 0.051 | 5.023 | 3.564 | 1.301 | 12.554 | 1.954 |
| $\varepsilon = 5$ | | 3.95 | 3.25 | 1.4 | | |
| $\varepsilon = 10$ | | 2.2 | 2.5 | 1.43 | | |

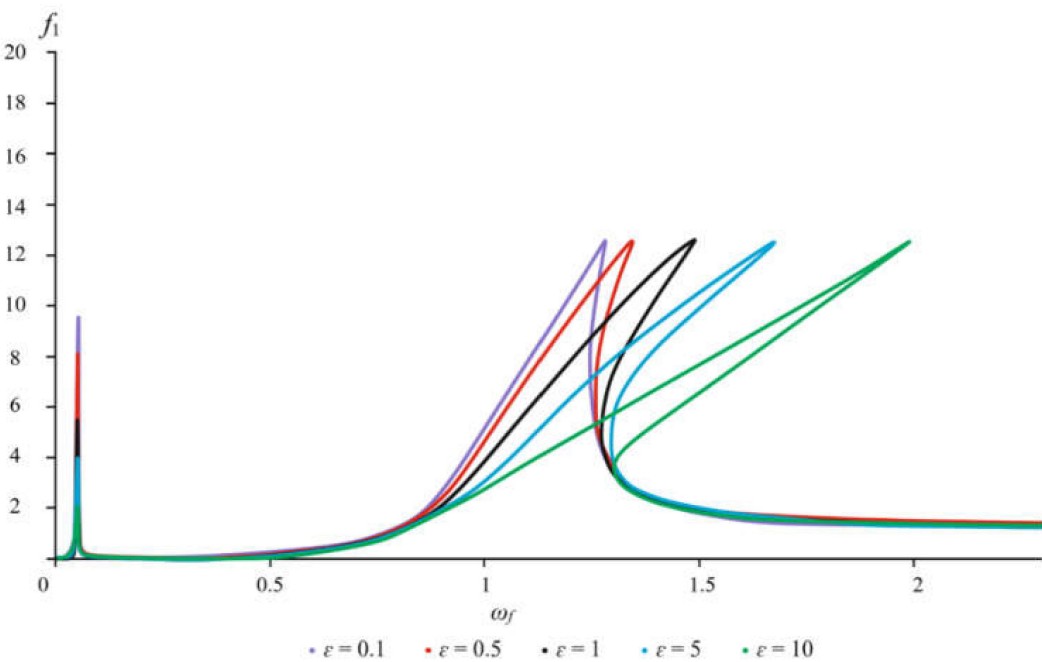

**Figure 5.** The amplitude-frequency characteristic of the rotor at various degrees of non-linearity.

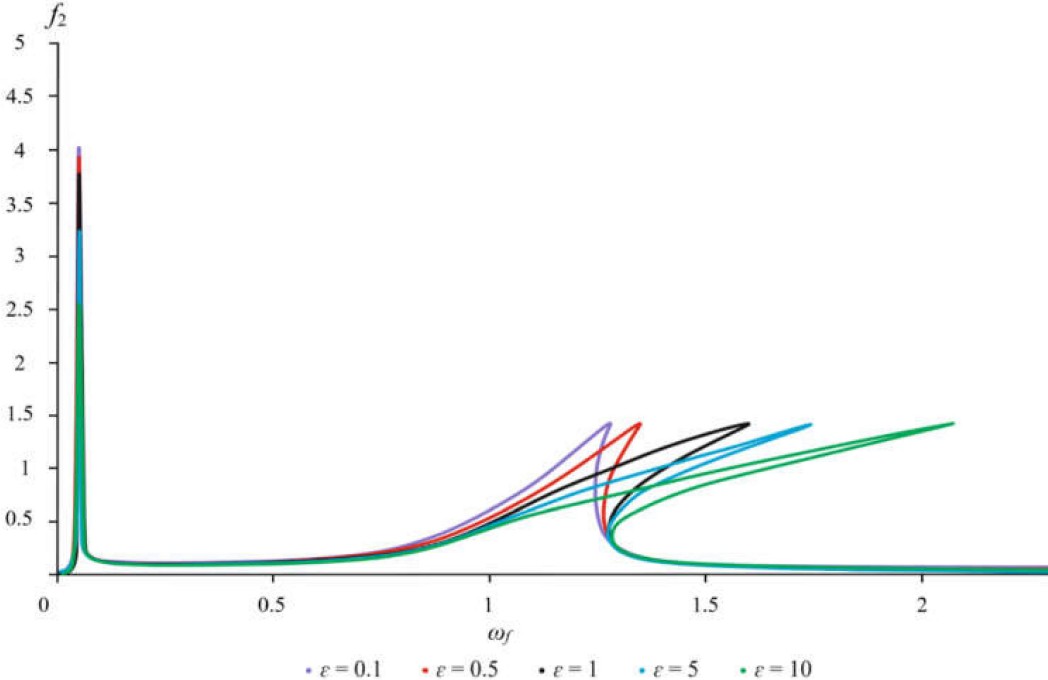

**Figure 6.** The amplitude-frequency characteristic of the foundation at various degrees of non-linearity.

In the case of an increase in the damping coefficients of the rotor and the foundation (Figures 7 and 8: cases $\zeta_1 = 5$ and $\zeta_2 = 5$, $\zeta_1 = 10$ and $\zeta_2 = 10$), the amplitudes for both the first and the second resonant regions decrease. For example, at $\zeta_1 = 5$ and $\zeta_2 = 5$, the foundation and rotor amplitudes at the first and second resonances decrease by a factor of 1.5. In the case of an increase in damping by one order of magnitude, i.e., $\zeta_1 = 10$ and $\zeta_2 = 10$, the amplitudes of the foundation and the rotor at the first and second resonances already decrease by a factor of 2.15. Shifts in the region of both the first resonant frequencies and the second ones are not observed in these cases. As the damping decreases

(Figures 7 and 8: cases $\zeta_1 = 0.1$ and $\zeta_2 = 0.1$, $\zeta_1 = 0.5$ and $\zeta_2 = 0.5$), the amplitudes of the rotor and foundation increase correspondingly in the first and second resonant regions. For example, when the damping coefficient of the rotor and foundation is halved, i.e., at $\zeta_1 = 0.5$ and $\zeta_2 = 0.5$, an increase in their amplitudes by a factor of 1.5 is observed. In the case of damping decrease by one order of magnitude, i.e., at $\zeta_1 = 0.1$ and $\zeta_2 = 0.1$, the foundation and rotor amplitudes already increase by a factor of 2.25. No shifts of the resonant regions are observed in these cases either. It is seen that an increase in the damping coefficient of the supports generally has a good effect on the dynamics of the system, however, due to the presence of structural limitations and the risk of shock loads at large values of the coefficient, the variation in the damping coefficients should be limited. The frequency and amplitude values are presented in more detail in Figures 7 and 8, and in Table 4.

**Table 4.** Amplitudes and resonant frequencies for different values of the damping coefficient.

| Parameter Value | First Resonance | | | Second Resonance | | |
|---|---|---|---|---|---|---|
| | Frequency $-\omega_f$ | Rotor's Amplitude-$f_1$ | Foundation's Amplitude-$f_2$ | Frequency $-\omega_f$ | Rotor's Amplitude-$f_1$ | Foundation's Amplitude-$f_2$ |
| $\zeta_1 = \zeta_2 = 0.1$ | | 11.25 | 7.87 | | 15 | 2 |
| $\zeta_1 = \zeta_2 = 0.5$ | | 7.5 | 5.25 | | 14 | 1.75 |
| $\zeta_1 = \zeta_2 = 1$ | 0.051 | 5.023 | 3.564 | 1.301 | 12.554 | 1.954 |
| $\zeta_1 = \zeta_2 = 5$ | | 3.33 | 2.33 | | 12 | 1.25 |
| $\zeta_1 = \zeta_2 = 10$ | | 2.325 | 1.63 | | 10.05 | 0.95 |

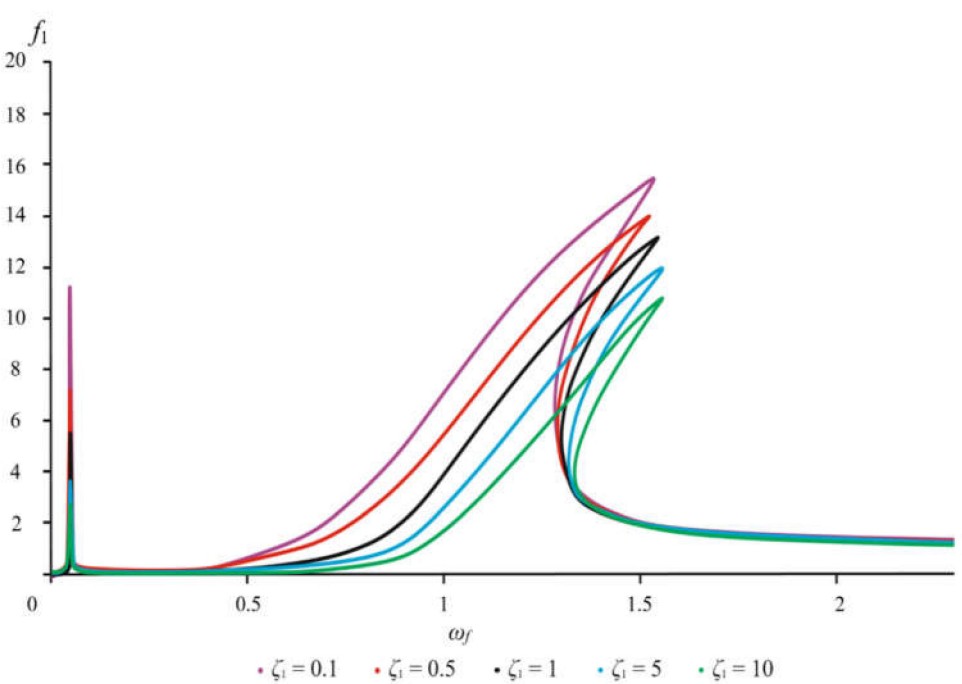

**Figure 7.** The amplitude-frequency characteristic of the rotor at different values of damping coefficients.

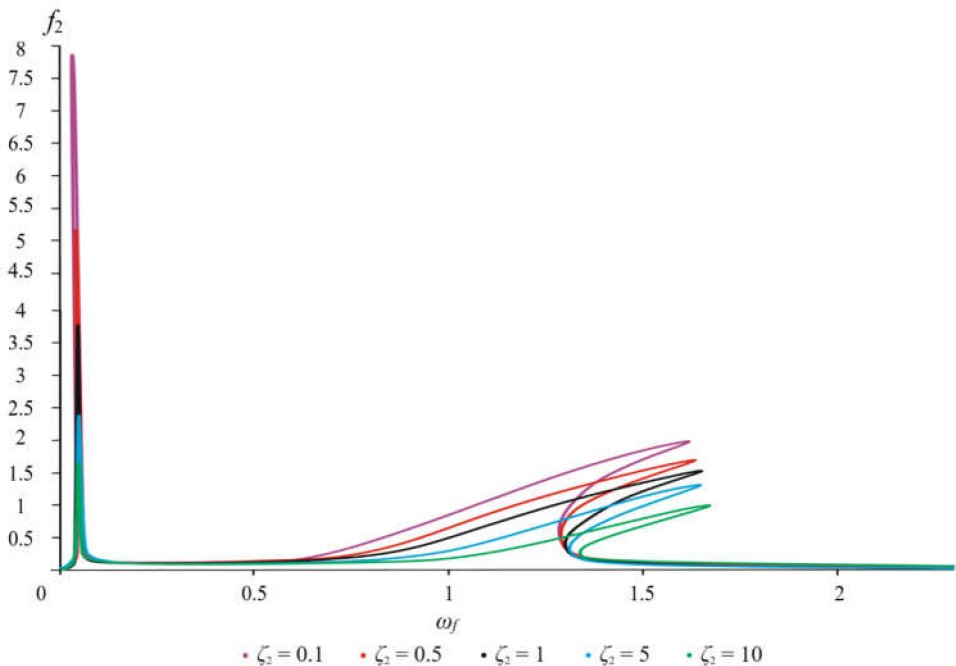

**Figure 8.** The amplitude-frequency characteristic of the foundation at different values of damping coefficients.

With an increase in the mass of the rotor (Figures 9 and 10: cases $\mu = 10$ and $\mu = 5$), an increase in the amplitudes of the rotor and foundation is observed in all areas of resonant frequencies. In the case of a fivefold increase in the mass of the rotor, i.e., at $\mu = 5$, the rotor amplitude increases by a factor of 1.4, and the foundation amplitude by a factor of 1.1. With a further increase in the mass ratio, the amplitudes also increase, for example, at $\mu = 10$, the difference in the amplitude ratios with the case of $\mu = 1$ already reaches 2.25 times. With an increase in the mass of the foundation (Figures 9 and 10: cases $\mu = 0.5$ and $\mu = 0.1$), in all cases and in all areas of resonant frequencies, a decrease in the amplitude values is observed, since in these cases the foundation acts as an anti-weight and dampens the oscillations of the system as a whole. For example, the values of the amplitudes at $\mu = 0.5$ for the rotor decrease in magnitude by 1.06 times, and for the foundation by 1.27 times. With a further increase in the mass of the foundation, the decrease in the amplitudes is more intense and is already less than for the case $\mu = 1$ by 2 and 1.97 times. The values of frequencies and amplitudes for different values of the mass ratio parameter are presented in more detail in Table 5 and in Figures 9 and 10.

**Table 5.** Amplitudes and resonant frequencies for different mass ratios.

| Parameter Value | First Resonance | | | Second Resonance | | |
|---|---|---|---|---|---|---|
| | Frequency -$\omega_f$ | Rotor's Amplitude-$f_1$ | Foundation's Amplitude-$f_2$ | Frequency -$\omega_f$ | Rotor's Amplitude-$f_1$ | Foundation's Amplitude-$f_2$ |
| $\mu = 0.1$ | | 2.5 | 2 | | 5.87 | 0.8 |
| $\mu = 0.5$ | | 4.7 | 2.75 | | 8.5 | 1.01 |
| $\mu = 1$ | 0.051 | 5.023 | 3.564 | 1.301 | 12.554 | 1.3 |
| $\mu = 5$ | | 7 | 3.9 | | 15.25 | 1.8 |
| $\mu = 10$ | | 11.25 | 4.25 | | 18.15 | 2.90 |

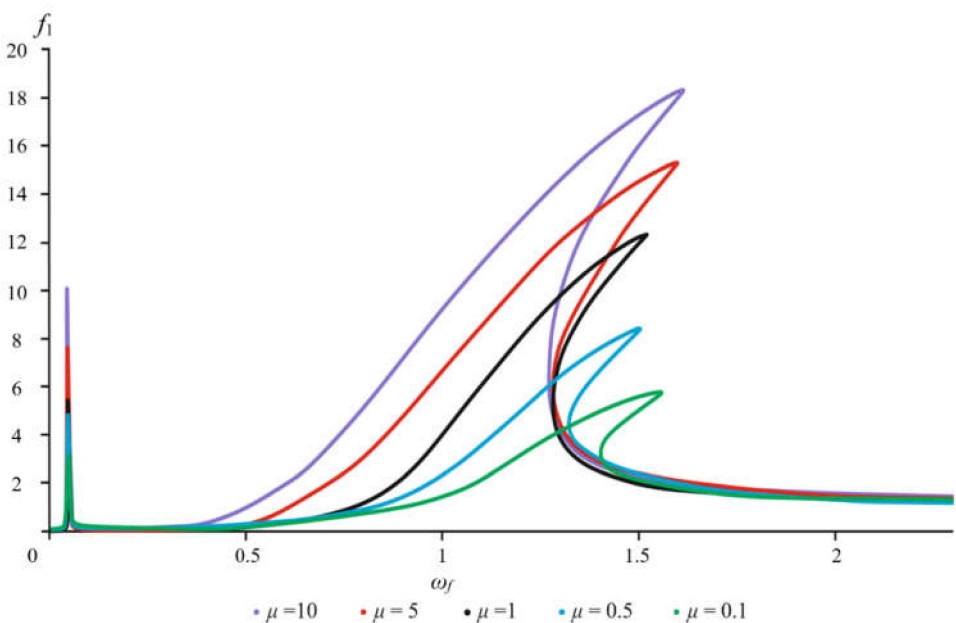

**Figure 9.** The amplitude-frequency characteristic of the rotor for different mass ratios.

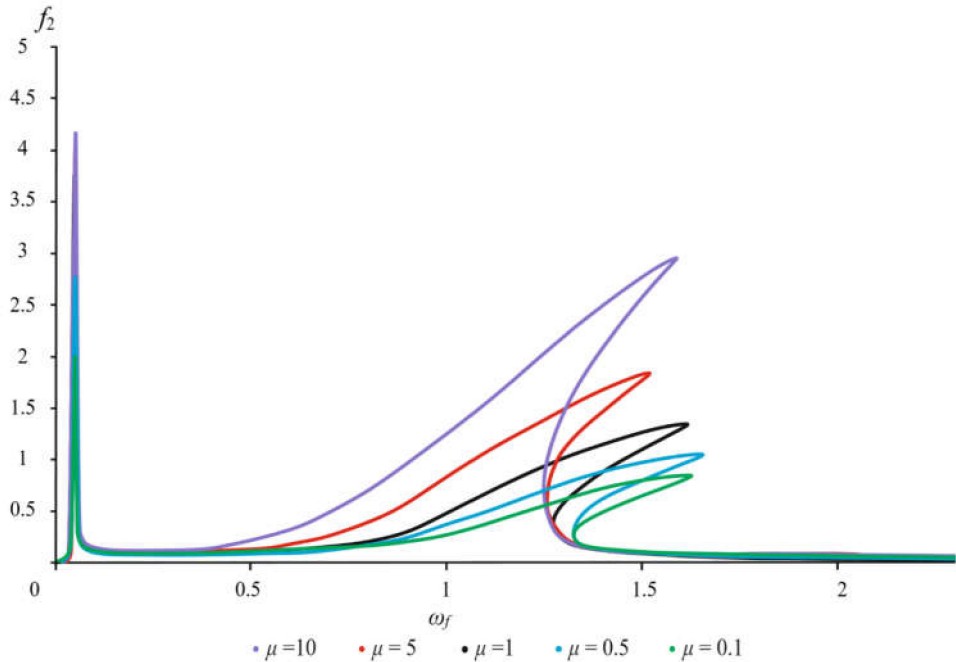

**Figure 10.** The amplitude-frequency characteristic of the foundation for different mass ratios.

When varying the linear stiffness coefficient λ, both a change in the magnitude of the amplitudes and their shift along the frequency axis are observed. With an increase in the coefficient $c_0$ (Figures 11 and 12: cases λ = 0.1 and λ = 0.5), the first and second resonant frequency regions shift to the left along the frequency axis, towards its decrease. In the case when λ = 0.5, the first region of resonant frequencies shifts to the value of $\omega_f = 0.04$, while the drop in amplitudes characteristic of the second region of resonant frequencies is already observed at $\omega_f = 1.25$. With a further decrease in the linear stiffness parameter, i.e., at λ = 0.1, the first resonant region already sets in at $\omega_f = 0.035$, and the drop in amplitudes is already observed at $\omega_f = 1.15$. As the stiffness coefficient $c_0$ decreases, the reverse picture is observed. For example, at λ = 5, the first region of resonant frequencies shifts in the

direction of higher frequencies to the value of $\omega_f = 0.06$, and the amplitudes drop already at $\omega_f = 1.4$. In the case of a further decrease in this coefficient, the first resonance occurs at $\omega_f = 0.07$, and the breakdown of the amplitudes occurs at $\omega_f = 1.45$. More detailed data on the values of frequencies and amplitudes are presented in Figures 11 and 12.

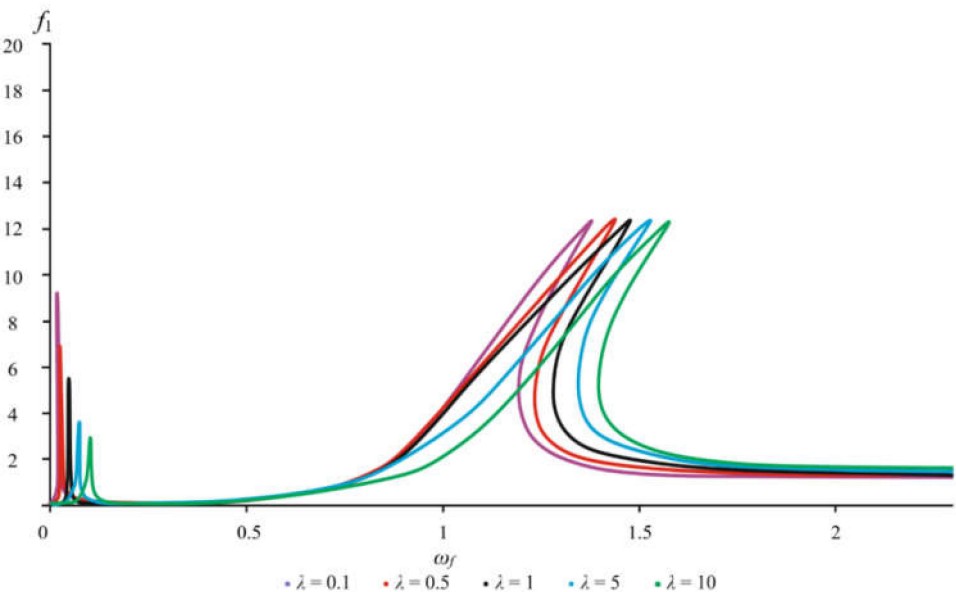

**Figure 11.** The amplitude-frequency characteristic of the rotor for different ratios of stiffness coefficients.

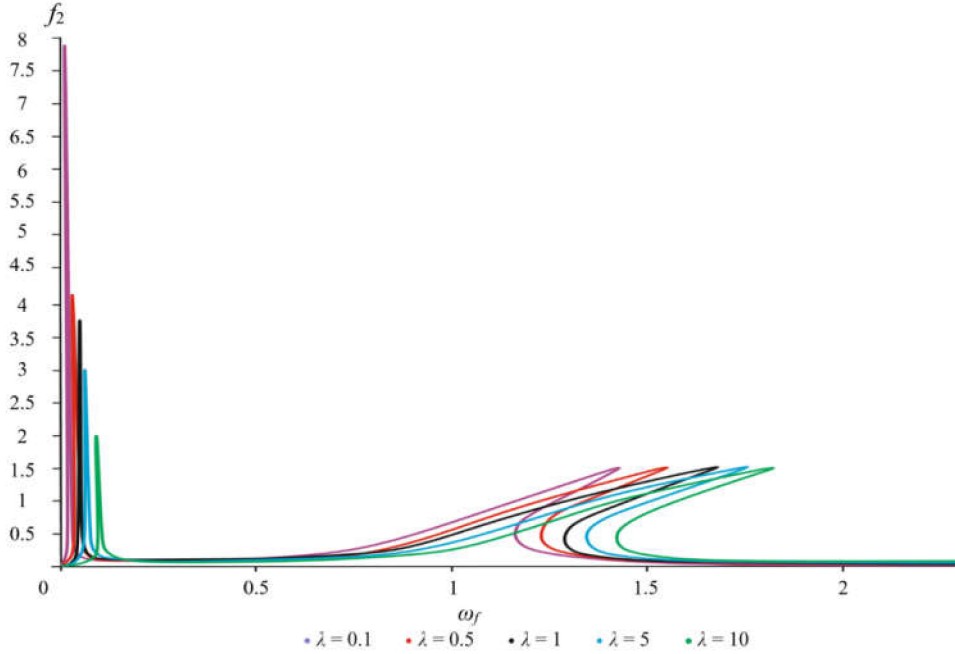

**Figure 12.** The amplitude-frequency characteristic of the foundation for different ratios of stiffness coefficients.

For high-speed rotor systems, when designing a rigid rotor rotating in elastic bearings, in order to reduce the amplitudes and vibration overloads, its first and second critical speeds are usually below the operating speeds, whereas in the operating frequency zone the self-centering effect is used. In our case, as the range of operating speeds is proposed to take the frequency interval between the first and second resonance, the region between the second

nonlinear and ultraharmonic resonance, or the interval lying behind the ultraharmonic resonance. In these areas, due to the self-centering effect, there is a significant decrease in the magnitudes of the amplitudes.

## 4. Conclusions

Thus, for optimal operation, it is necessary that in the stationary mode the operating speed of the system be in the interval between the first linear and the second nonlinear resonance, i.e., $0.051 < \omega_f < 1.301$ (the first working region), or between the second and ultraharmonic resonances, i.e., $1.301 < \omega_f < 3$ (the second working region), or in the region behind the ultraharmonic resonance (the third working region), where, due to the self-centering effect, the values of the rotor and foundation amplitudes decrease.

An increase in the nonlinearity parameter up to $\varepsilon = 10$ has a positive effect on the amplitude values. In this case, it should be taken into account that the frequency of the amplitude breakdown is shifted towards higher rotor speeds.

The operation of the system is also positively affected by an increase in the damping coefficients ($\zeta_1$, $\zeta_2$) and an increase in the mass of the foundation, as in this case the foundation acts as an anti-load and contributes to vibration damping.

An increase in the linear rigidity parameter $\lambda$ leads to a decrease in the first resonance and a shift in the second resonance, which also contributes to a smoother transition through the resonant frequencies.

In all of the above cases, it should be taken into account that the limit of the maximum values of parameters of nonlinearity, damping, foundation mass and linear rigidity is limited by the design and technical features of a particular industrial centrifuge.

A technique that enables us to calculate the maximum value of the amplitudes and construct the frequency characteristics of forced oscillations of the "rotor-foundation" system on rolling bearings with a nonlinear characteristic based on Jacobi elliptic functions has been developed. The optimal parameters associated with the mass of the foundation, the coefficients of linear and nonlinear stiffness, as well as the damping coefficients, at which the magnitudes of the amplitudes have optimal values without the need for fundamental structural changes, are determined. The specific features of linear and nonlinear behavior of the system with many degrees of freedom caused by foundation vibrations are considered.

**Author Contributions:** Conceptualization, A.K.; methodology, A.K.; software, G.-G.A.I.; validation, A.Z. and A.K.; formal analysis, A.K. and A.Z.; investigation, A.K. and A.Z.; resources, A.K. and A.Z.; data curation, G.-G.A.I., A.Z. and A.K.; writing—original draft preparation, A.K.; writing—review and editing, A.Z.; visualization, G.-G.A.I.; supervision, A.Z.; project administration, A.K.; funding acquisition, A.K. All authors have read and agreed to the published version of the manuscript.

**Funding:** This work has been supported financially by the research project AP08856167 of the Ministry of Education and Science of the Republic of Kazakhstan and was performed at Research Institute of Mathematics and Mechanics in Al-Farabi Kazakh National University, which is gratefully acknowledged by the authors.

**Institutional Review Board Statement:** Not applicable.

**Informed Consent Statement:** Not applicable.

**Data Availability Statement:** Data sharing not applicable to this article as no datasets were generated or analysed during the current study.

**Conflicts of Interest:** The authors declare no conflict of interest.

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
