# Peer review of "Investigation of Nonlinear Forced Vibrations of the “Rotor-Movable Foundation” System on Rolling Bearings by the Jacobi Elliptic Functions Method"

_applsci, doi:10.3390/app12147042_

Round 1
Author Response
We have written detailed answers to all questions in accordance with the opinions of reviewers. We have also attached an individual correction file with a detailed description of the changes

Reviewer 2 Report
The paper has an interesting topic but cannot be published in the present form. There are several things appearing in the paper that are difficult to understand.
First of all, the quotations in the Introduction do not coincide with what is written in the References. The writing is negligent: missing numbering of formulas, errors in considering the notations (see F_C in formula and F_r in the row 109, or “cubic and linear coefficients” on the row 114).
My most serious concerns are related to the fact that it is not clear underlined what novelty the paper is bringing. The authors should explain what is the connection of their paper with another one, “Applications of Jacobi’s Elliptic Functions in Forced Vibration in Nonlinear Rotor Dynamics”, with a different authorship (Gulama-Garip Alisher Ibrayev, Isaac Elishakoff , Almatbek Kydyrbekuly). It is not cited and I did not understand if it was published elsewhere. Anyway, it seems that the References in the paper we analyze here, references wrong cited as I mentioned, are taken from this last paper.
Author Response

(The authors gave the same response as above.)

Reviewer 3 Report
In the work, a technique that enables us to calculate the maximum value of the amplitudes and construct the frequency characteristics of forced oscillations of the “rotor-foundation” system on rolling bearings with a nonlinear characteristic based on Jacobi elliptic functions is developed. There are some detailed questions and suggestions for this work are expected to be addressed properly.
1. There are some errors in the format of this paper. Many formulas have no labels, and some paragraphs have no indentation at the beginning, which causes great reading barriers.
2. Lines 172 to 186 in this paper are highly identical with section 2.1 in reference [35], and even Figure 2 in this paper is the same as Figure 1 in reference [35]. I think it is inappropriate to repeat the content in reference.
3. The data in the column "Rotor's Amplitude-f1" in Table 1 are inconsistent with that shown in Figure 3.
4. In parameter analysis, why choose the five values of 0.1, 0.5, 1, 5 and 10?
5. It is mentioned in the conclusion that "The optimal parameters associated with the mass of the foundation, the coefficients of linear and nonlinear stiffness, as well as the damping coefficients", but the optimal value range of these four parameters is not clearly proposed.
Author Response

(The authors gave the same response as above.)

Round 2
Reviewer 2 Report
Despite this second version of the paper in the form I got it has many misprinting, I guess it is a technical problem only and I note that authors answered to my concerns expressed in the first review.
As I mentioned there, the paper is interesting and can be published now.
Reviewer 3 Report
Good with the revision.